# Building a Cardiovascular Disease Prediction Model for Smartwatch Users Using Machine Learning: Based on the Korea National Health and Nutrition Examination Survey

**DOI:** 10.3390/bios11070228

**Published:** 2021-07-08

**Authors:** Min-Jeong Kim

**Affiliations:** Department of Consumer Economics & Design Research Institute for Creativity and Convergence, Sookmyung Women’s University, Seoul 04310, Korea; min-jeong.kim@sookmyung.ac.kr; Tel.: +82-2-2077-7818

**Keywords:** smartwatch, cardiovascular disease prediction model, machine learning, logistic regression, artificial neural network, support vector machines, Korea National Health and Nutrition Examination Survey

## Abstract

Smartwatches have the potential to support health care in everyday life by supporting self-monitoring of health conditions and personal activities. This paper aims to develop a model that predicts the prevalence of cardiovascular disease using health-related data that can be easily measured by smartwatch users. To this end, the data corresponding to the health-related data variables provided by the smartwatch are selected from the Korea National Health and Nutrition Examination Survey. To classify the prevalence of cardiovascular disease with these selected variables, we apply logistic regression, artificial neural network, and support vector machine among machine learning classification techniques, and compare the appropriateness of the algorithm through classification performance indicators. The prediction model using support vector machine showed the highest accuracy. Next, we analyze which structures or parameters of the support vector machine contribute to increasing accuracy and derive the importance of input variables. Since it is very important to diagnose cardiovascular disease early correctly, we expect that this model will be very useful if there is a tool to predict whether cardiovascular disease develops or not.

## 1. Introduction

As modern society enters an aging society, health problems caused by chronic diseases are intensifying. In Korea, deaths from chronic diseases account for more than 50% of all deaths. According to Statistics Korea’s 2019 report on the cause of death in Korea, the number one cause of death was cancer (27.5%), and the second place was heart disease (10.5%), the 3rd place was pneumonia (7.9%), the 4th place was cerebrovascular disease (7.3%), with cardiovascular disease accounting for a total of 17.8% [1].

Absolute risk levels of cardiovascular disease rise progressively with age, even in the absence of risk factors, but risk factors for cardiovascular disease differ by age group [2]. The prevalence of cardiovascular disease among young people is lower than that of other age groups because physiological risk factors such as lipid metabolism and vascular status in young people do not have a significant effect on cardiovascular disease. The characteristics of the risk factors for cardiovascular disease in the middle aged were gradually affected by physiological risk factors compared to the young, but it was found that it took time for it to deteriorate significantly. Therefore, the middle aged are only classified as a potential risk group for cardiovascular disease and are not receiving much attention compared to the actual risk group, the elderly [2]. This means that if the middle aged, who are the current potential risk group, continue to have an unhealthy lifestyle that they have had since their youth, they are very likely to become a high-risk group of cardiovascular disease [3]. Therefore, it is very important to predict and prevent a person at risk of cardiovascular disease in advance, and it will be very useful if there is a tool to predict the onset of cardiovascular disease.

In the modern society, social interest in health is remarkable, and with the advent of the fourth industrial revolution, various solutions combining personal health information and artificial intelligence are being developed, which can predict future diseases through analysis of biometric and health-related data [4]. In Korea, the Ministry of Health and Welfare is establishing an infrastructure by establishing a cardiovascular disease management plan, and accordingly, personal health-related data are also increasing every year [5]. Integration of clinical decision support with such computer-based patient records could reduce medical errors, enhance patient safety, decrease unwanted practice variation, and improve patient outcome [6].

Smartwatches have the potential to support health in everyday living by enabling self-monitoring of personal activity, obtaining feedback based on activity measures, allowing for in situ surveys to identify patterns of behavior, and supporting bi-directional communication with health care providers and family members [7]. Samsung Electronics has already released the Galaxy Watch, which can measure and record blood pressure and electrocardiogram in 2020. The Samsung Health Monitor application received the Ministry of Food and Drug Safety (MFDS) clearance for blood pressure and electrocardiogram measurement in Korea in May 2020 [8] and received a Conformity to Europe (CE)-marking in December 2020 [9]. A CE-marking is an administrative sign to highlight that a product complies with EU safety, health and environmental requirements. Samsung Electronics is also planning to release a smartwatch with a blood glucose measurement function newly applied in the second half of 2021 [10].

This paper aims to develop a model that predicts the prevalence of cardiovascular disease using health-related data that can be easily measured by smartwatch users. To this end, the data corresponding to the health-related data variables provided by the smartwatch are selected from the Korea National Health and Nutrition Examination Survey published annually by Korea Disease Control and Prevention Agency. To classify the prevalence of cardiovascular disease with these selected variables, we apply logistic regression, artificial neural network, and support vector machine among machine learning classification techniques, and compare the appropriateness of the algorithm through classification performance indicators. Next, after evaluating the accuracy of the three generated models, we select the optimal model with the highest accuracy. Further, we analyze the change in accuracy by changing arguments of the selected model and derive the importance of input variables. Since it is very important to diagnose cardiovascular disease early correctly, we expect that this model will be very useful if there is a tool to predict whether cardiovascular disease develops or not.

## 2. Review of Literature

The Framingham Risk Score (FRS) was first developed based on data obtained from the Framingham Heart Study, to estimate the 10-year risk of developing coronary heart disease [11]. The purpose of calculating the FRS is to identify high-risk patients who need immediate attention and intervention, motivate patients who need treatment to reduce risk factors, and to reduce risk factors based on the overall risk assessment [2]. Since the Framingham Heart Study reported a cardiovascular disease risk prediction method in 1976, an increasing number of risk assessment tools have been developed to cardiovascular disease risk in various settings. The Framingham Heart study results are fundamental evidence for the prediction of cardiovascular disease risk. However, the clinical utility of a disease prediction model can be population-specific because the baseline disease risk, subtype distribution of the disease, and level of exposure to risk factors differ by region and ethnicity [12].

In Korea, the FRS was used for cardiovascular disease risk, but studies have been published that it is overestimated for Koreans [12,13]. There is also a limitation that the FRS is only predictable about the likelihood of coronary artery disease. Recently, the European cardiovascular disease risk assessment model, Systematic COronary Risk Evaluation (SCORE) and Atherosclerotic cardiovascular disease (ASCVD) risk assessment model are also used, but it is known that it is still overestimated for Asians [14]. Therefore, statistical analysis studies on the risk factors of cardiovascular disease in Korean adults were conducted using the Korea National Health and Nutrition Examination Survey [15,16], and studies on the risk factors of cardiovascular disease for patients in domestic hospitals were also conducted [17].

In recent years, the use of applications using mathematical models has been increasing in the field of healthcare, as data size has increased and the constraints on computational power have gradually disappeared. According to a study by Anand et al. [18], between 2003 and 2015, studies using data mining in the field of healthcare were increasing. Among them, 51.5% of studies using data mining and 39.3% of studies using machine learning methods were reported. As such, there are many studies on disease prediction using data mining [19,20,21]. Among them, studies on predicting cardiovascular disease are also being conducted [4,22]. However, these studies predict cardiovascular disease using data attributes that are difficult to measure if ordinary people do not go to the hospital. In this study, we intend to build a cardiovascular disease prediction model by considering only data that can be measured by ordinary people using smartwatches.

## 3. Research Methodology

### 3.1. Data Source

This study utilized the Korea National Health and Nutrition Examination Survey on data attributes corresponding to health-related data provided by smartwatches. The Korea National Health and Nutrition Examination Survey is conducted to determine the health and nutrition status of the people in accordance with Article 16 of the National Health Promotion Act, and the purpose of this survey is to produce statistics that are representative and reliable at the national and provincial level on health behavior, chronic disease, prevalence, food and nutrition. The final 6170 people were selected for analysis of this study after removing missing and abnormal values from the latest 2019 survey results. The total data was a survey of 8110 respondents, but there were 402 missing values in prevalence questions of six disease, 505 missing values from the actual health-related data, and 1033 respondents chose ‘don’t know/no answer’ for Likert scale responses. The final valid number of data was derived as 6170.

### 3.2. Variables

According to the Korea National Health and Nutrition Examination Survey, respondents are currently checking the prevalence of hypertension, dyslipidemia, stroke, myocardial infarction, angina pectoris, and diabetes. Therefore, the target variable of this study was coded as ‘0’ for those who answered that they had no disease for all 6 diseases, and ‘1’ for those who answered that there was a prevalence of at least one disease. The reason for linking the prevalence of diabetes and the presence of cardiovascular disease is the result of research showing that if blood sugar continues to be high due to diabetes, blood vessels are damaged and various cardiovascular diseases accompany this [23]. In addition, dyslipidemia was considered because it is the main cause of cardiovascular disease with increased total cholesterol, Low-Density Lipoprotein-Cholesterol (LDL-C), and triglycerides or decreased High-Density Lipoprotein-Cholesterol (HDL-C) in the blood [24].

The input variables used in this study were selected in consideration of functions that can be measured in Samsung Electronics’ smartwatches. As mentioned above, Samsung Electronics’ smartwatch can measure blood pressure, and electrocardiogram, and blood glucose can be measured in the second half of 2021. The smartwatch is already providing heart rate, oxygen saturation, and stress index functions. The stress index indicates the current level of stress by analyzing breathing, heart rate, and body temperature for 10 to 20 s by pressing the measurement button while wearing the smart watch on the wrist. The blood glucose measurement function has not yet been released, but this study considered it as an input variable because it will be included in the Samsung Electronics’ smartwatch that will be released soon in 2021. The screen of the Samsung Health Monitor application linked to the Samsung Electronics’ smartwatch is shown in Figure 1.

Finally, the input variables used in the Korea National Health and Nutrition Examination Survey are as follows: First, it included gender and age reflecting the demographic levels. This is data that can be entered into the prediction model without any measurement because it is the personal information of a smartwatch user. Next, as mentioned above, the functions provided by Samsung Electronics’ smartwatch such as systolic blood pressure (sbp), diastolic blood pressure (dbp), fasting blood glucose, pulse rate for 15 s, and perceived stress were added. The level of perceived stress is classified into 1 = feeling too much, 2 = feeling much, 3 = feeling a little, and 4 = feeling little on a Likert 4-point scale the Korea National Health and Nutrition Examination Survey. In this study, the perceived stress is used instead of the smartwatch’s stress index, based on a study comparing computer-based heart rate variability (HRV) and self-reporting scales (SRS), known as subjective evaluation methods, to evaluate the treatment effectiveness on emotional improvement of depressed patients [25]. However, in Song et al.’s study [25], the SRS was conducted with values that included four test areas: mental stress scale (MSS), physical stress scale (PSS), Beck anxiety inventory (BAI), and Beck depression inventory (BDI), whereas our variable consisted of only one subjectively measured Likert 4-point scale. Finally, self-awareness of body shape and Body Mass Index (BMI) were included. Self-awareness of body shape is classified as thin (1 = very thin, 2 = slightly thin), 3 = moderate, and obese (4 = slightly obese, 5 = very obese) on a Likert 5-point scale the Korea National Health and Nutrition Examination Survey, and BMI is data that can be calculated simply by entering height and weight. Input variables were used without modification in the attributes of variables provided by the Korea National Health and Nutrition Examination Survey. The descriptions of variables used in this study are shown in Table 1.

### 3.3. Research Procedure

The procedure of the study to develop a cardiovascular disease prediction model, which is the purpose of this study, is as follows. In the 2019 data of the Korea National Health and Nutrition Examination Survey, the prevalence of cardiovascular disease, which is the target variable, was newly created using the prevalence or not of diseases related to cardiovascular disease, and nine input variables were selected as shown in Table 1. Next, we eliminate missing and abnormal values through the data preprocessing process and apply three machine learning methods to the final data (6170 respondents) for generating a cardiovascular disease prediction model. In order to avoid the problem of overfitting the model, about 70% (4319 respondents) of the observations in the entire data set are trained as a training data set. Additionally, the performance of the optimal model is verified with the test data set of 30% (1851 respondents), which is the remaining data set not used for model training. This is to check how accurately the trained model performs prediction or estimation given new data. Table 2 shows the variables’ descriptive statistics of training and test data sets. Next, after evaluating the accuracy of the generated models, we select the optimal model. Further analysis is then performed by varying the arguments for the selected model.

In this study, logistic regression, artificial neural network, and support vector machine are used as machine learning methods to generate the cardiovascular disease prediction model. Logistic regression analysis is a widely used method in data in which the type of the target variable is categorized and dichotomous, and has been used as a model for classification and prediction in various fields including healthcare [26]. Artificial neural network is a method of creating several layers between the input node and the output node, comparing the output node value and the target value through a calculation process, and then repeatedly performing it until the nearest value is derived [27]. Artificial neural network has the advantage of being able to produce appropriate results by applying weights even with incomplete inputs, but has a disadvantage in that it is difficult to explain the rules due to the presence of a hidden layer. Support vector machine is a classification or prediction method by finding a separating boundary called hyperplane [28]. The main advantage of support vector machine is that it can, with relative ease, overcome ‘the high dimensionality problem’, i.e., the problem that arises when there are a large number of input variables relative to the number of available observations. This technique has recently been used to improve methods for detecting diseases in clinical settings [29,30]. Moreover, support vector machine has demonstrated high performance in solving classification problems in bioinformatics [31]. We use R-4.0.2 program for conducting modeling and statistical analysis of three machine learning methods.

## 4. Results

### 4.1. Performance Evaluation Index

To evaluate the robustness of the estimates from the cardiovascular disease prediction models, we use four statistics including Accuracy, Recall, Precision, and F1 score. First, Accuracy is the number of correctly predicted data divided by the total number of data, and the formula is as follows.
$$\mathrm{Accuracy}=\frac{\mathrm{True}\;\mathrm{Positives}+\mathrm{True}\;\mathrm{Negatives}}{\mathrm{True}\;\mathrm{Positives}+\mathrm{True}\;\mathrm{Negatives}+\mathrm{False}\;\mathrm{Positives}+\mathrm{False}\;\mathrm{Negatives}}$$

Next is Recall, which means the number of data that the model recognized as true among data that are actually true, and the formula is as follows.
$$\mathrm{Recall}=\frac{\mathrm{True}\;\mathrm{Positives}}{\mathrm{True}\;\mathrm{Positives}+\mathrm{False}\;\mathrm{Negatives}}$$

Third, Precision is the number of data that are actually true among the data predicted by the model as True, and is a measure that has a trade-off relationship with Recall. The formula is as follows.
$$\mathrm{Precision}=\frac{\mathrm{True}\;\mathrm{Positives}}{\mathrm{True}\;\mathrm{Positives}+\mathrm{False}\;\mathrm{Positives}}$$

Finally, the F1 score is an index that can explain how effective the model is in addition to precision and recall in measuring the performance of the model, and is the harmonic mean of precision and recall. The F1 score formula is as follows.
$$F1\;\mathrm{score}=2\times \frac{\mathrm{Precision}\;\times \mathrm{Recall}}{\mathrm{Precision}+\mathrm{Recall}}$$

### 4.2. Performance Comparison of Models

The performance comparison results of three machine learning methods for cardiovascular disease prediction model on 1851 test data set, 30% of the total 6170 observational data are shown in Table 3. In this paper, we use the “caret” package [32] to create a 70/30% split of the final data. In addition, the logistic regression analysis is implemented using the “glm” function of the R-4.0.2 program, the “neuralnet” package [33] for the artificial neural network, and the “e1071” package [34] for the support vector machine. For the primary comparative analysis, all three methods use default values for the arguments in the classification model, and in the artificial neural network, we set to have two hidden layers and six hidden nodes for each hidden layer.

As can be seen from these results, among the three models, support vector machine has the highest Accuracy, Recall, and F1 Score. In Precision, logistic regression and artificial neural network are higher than support vector machine, but support vector machine showed higher performance on three other scales. This is a similar result to previous research [31]. Therefore, in the next section, we will further analyze what structure or parameters of the support vector machine contribute to increasing accuracy.

### 4.3. Support Vector Machine

#### 4.3.1. Optimal Model of Support Vector Machine

In the previous section, three machine learning methods were applied without normalizing for the nine input variables. As a result, it was found that the performance of support vector machine was the best. As a first step in further analysis of the support vector machine model, we normalized the data and then applied support vector machine again. Values of input variables were normalized to values from −1 to +1. The results showed the same classification performance (Accuracy 83.04%) as without normalization. Further analysis in the next step seeks to compare performance while varying the penalty parameter C, kernel parameter Gamma, and kernel function, which are parameters that affect the performance of support vector machine [35]. In this paper, we select the optimal pair (C, Gamma) from C = {1(default),10,100} and Gamma = {0.01, 0.1, 1/number of input variables(default)}. Additionally, different kernel functions, including linear, polynomial, radial basis functions (RBF), and sigmoid were tested and selected for the models on the basis of performance. In the case of the linear kernel function, the Gamma value is not required, so the search is performed for a total of 30 combinations. As a performance evaluation method, we compare with accuracy and area under the curve (AUC). Table 4 shows the classification accuracy and AUC obtained from the four kernel functions using default values C and Gamma. The highest accuracy was 0.8304 in the RBF kernel, followed by the linear kernel. However, the AUC value of the linear kernel is slightly higher than that of the RBF kernel.

Table 5 shows the classification accuracy and AUC obtained according to the optimal C and gamma values in the four kernel functions. The results also showed the highest accuracy of 0.8320 at C = 10 and Gamma = 0.01 in the RBF kernel and the highest AUC of 0.781 at C = 1 in linear kernel.

Figure 2 is a graph of receiver operating characteristic (ROC) curve according to the optimal C and gamma values of the four kernel functions obtained in Table 5. The ROC curve is a two-dimensional measure of classification performance in which the false positive rate is the *x*-axis, and the true positive rate is the *y*-axis. It can be understood as a plot of the probability of correctly classifying the positive examples against the rate of incorrectly classifying true negative examples. The performance of ROC curve improves the further the curve is near to the upper left corner of the plot [36]. The area under the ROC curve is called AUC, if one model’s AUC is larger than the other’s, it is on average a better model [37]. As shown in Figure 2, except for polynomial, the other three kernel functions show almost similar ROC curve.

#### 4.3.2. Features’ Importance

Then, we investigate which input variables are of high importance for predicting cardiovascular disease, which is the purpose of this study. Support vector machine with linear kernel is possible to access the classifier coefficients. These weights figure the orthogonal vector coordinates orthogonal to the hyperplane. Feature importance can, therefore, be determined by comparing the size of these coefficients to each other. In this paper, we use the “varImp” function of the “caret” package [32] to calculate features’ importance. The importance result for the nine input variables is shown in Figure 3. First, the age (0.8652) was shown as the variable with the highest ranking. It is consistent with previous research that shows that the absolute risk level for cardiovascular disease rises gradually with age [2]. Next, systolic blood pressure (HE_sbp) (0.7386), fasting blood glucose (HE_glu) (0.7374), and body mass index (HE_BMI) (0.6404) were in the upper range, diastolic blood pressure (HE_dbp) (0.5435), self-awareness of body shape (BO1) (0.5420), perceived stress (BP1) (0.5260), and pulse rate for 15 s (HE_PLS) (0.5170) appeared in the following order. The result of the effect of blood pressure on cardiovascular disease is consistent with previous study showing that systolic blood pressure is superior to diastolic blood pressure as a predictor of cardiovascular events [38]. Finally, gender (0.5082) was placed in the lowest ranking.

From the above result, age was shown as the variable with the highest ranking and gender was placed in the lowest ranking. In order to control for these variables, we examine the importance of features by age group and gender again. First, the train data set is grouped by age intervals, we examined the importance of different features for each group. Figure 4 summarizes the results of features’ importance by age group. As can be seen from Figure 4, the importance of features was different for each age group. In particular, the change in importance priority was clear from the 20 s to the 40 s, and the feature that had the most influence on the prevalence of cardiovascular disease in the 20 s and 30 s was HE_dbp. This is consistent with the research [39], which shows that the risk of cardiovascular disease is high when the elderly have high sbp and young people have high dbp as well as sbp. On the other hand, after 40 s, HE_glu ranked highest in the priority of feature importance. These results are also consistent with the study that the prevalence of diabetes increases with age, and that people with diabetes have a higher risk of cardiovascular disease [40]. However, BP1 showed the lowest importance in certain age groups (30 s, 40 s, and 80 s) in Figure 4, which is different from the results in Figure 3. Although there are research results that psychosocial factors such as acute or chronic stress affect cardiovascular disease [41] and psychological stress reactivity is associated with atherogenesis in youth [42], research on factors affecting cardiovascular disease by age group has not yet been conducted. In addition, since the perceived stress of the Korea National Health and Nutrition Examination Survey used in this study is a Likert 4-point scale value that was subjectively checked by the respondents, it is considered that an analysis using a precise measurement value is necessary in future studies.

Next, we classify the training data set by gender and analyze the feature importance by gender. The results are summarized in Table 6 by ranking of the importance of the features. Age was the highest ranking feature for both male and female, but HE_glu in male and HE_sbp in female came next. This result is consistent with the findings of a Korean study that the prevalence of cardiovascular disease was higher in women with high sbp and higher in men with diabetes [43].

## 5. Discussion

In this paper, we applied three machine learning methods: logistic regression, artificial neural networks, and support vector machines to build a model for predicting cardiovascular diseases for smartwatch users based on their own health-related data. As a result of model performance evaluation, support vector machine showed the best performance. Accordingly, further analysis was conducted on support vector machine. In general, while applying support vector machine to classification problems, the selection for the parameters C and Gamma does not take into account the characteristics of the data, and the default values, that is, the number of C = 1 and gamma = 1/dimensions(number of input variables), are often used [44,45]. However, in this study, it was confirmed that the search for optimal parameters is important when applying support vector machine to classification problems because support vector machine can improve the model performance by setting parameters suitable for the characteristics of the data through experiments that change kernel functions and parameters. In this study, the optimal support vector machine model uses RBF kernel function, higher C values and smaller Gamma values than those of the basic model. Higher C values mean higher penalties for errors than the basic model. Therefore, the optimal support vector machine model with higher C values has a higher performance than the basic model, but has a disadvantage that it takes a lot of computation time as it uses high C value. As with parameter C, if Gamma is too low, there is a high probability of underfitting, and if Gamma is too high, there is a risk of overfitting. Therefore, it is necessary to find appropriate values for both parameters.

## 6. Conclusions

Until now, smartwatch users in Korea have been staying at the level of managing their health through health-related data such as heart rate, blood pressure, and weight [46]. However, as the number of smartwatch users increases and additional functions such as electrocardiogram and blood glucose are released on the smartwatch, this study was initiated under the assumption that it would be possible to predict the disease’s prevalence based on these data.

In this paper, nine input variables were selected from the Korea National Health and Nutrition Examination Survey 2019 data in consideration of functions that can be measured in Samsung Electronics’ smartwatches, and prevalence of cardiovascular disease, which is target variable, was generated using variables related to cardiovascular disease prevalence. We divided the entire data sets into training (70%) and test (30%). After that, three machine learning methods (logistic regression, artificial neural network, support vector machine) were used to classify cardiovascular disease on training data set and to check how accurately the trained model performs prediction given test data set. As a result of classification, support vector machine was the highest in terms of accuracy.

Subsequently, in the classification problem, further analysis of the support vector machine was performed by varying kernel functions and parameters (C and Gamma), assuming that kernel functions and parameters affect the performance of the model. The analysis results show that the optimal support vector machine model uses the RBF kernel function and has a higher C value and a lower Gamma value than those of the basic model. As such, machine learning methods can have different performance depending on parameters. Therefore, researchers will have to build an optimal model by controlling the parameters as closely as possible.

Finally, we examined the importance of input variables. Age was the highest, and systolic blood pressure, fasting blood glucose, and body mass index were ranked in the top, while gender was ranked the lowest. Following age, systolic blood pressure has been shown to be an important variable, which shows that these results are consistent with previous studies [2,38]. In addition, the importance of features was analyzed according to age groups and gender groups. As a result, the features’ importance of each age group showed a lot of difference in the ranking. In particular, there was a clear difference between the young and the elderly, which is similar to the results of previous studies [39,40]. According to the gender group, there was a difference in the 2nd and 3rd places in the importance of features, which is also the same as the Korean study results [43].

The health-related data measured on Samsung Electronics’ smartwatch has already received MFDS clearance [8] and CE marking [9], so it can be seen that there is no difference from the corresponding data of the Korea National Health and Nutrition Examination Survey. However, in the case of the stress index, it is difficult to say that it is sufficiently related to the result of the Korea National Health and Nutrition Examination Survey. Therefore, since the stress index variable is a limitation of this study, it is necessary to reconstruct the predictive model using a precise measurement value or quantification from an integrated perspective on the stress index in the future.

In this study, all respondents who answered that they had at least one of the six diseases known to be related to cardiovascular disease were included as patients with cardiovascular disease, and the study was conducted. However, each of the six diseases may show different characteristics and symptoms, which means that the influencing factors of the disease prediction model may be different. Therefore, it is thought that it is necessary to look at the differences between diseases by constructing a model that predicts each of the six diseases in future studies. It would be helpful to understand the distribution of false-negative subjects in each disease and which models misclassify subjects with the disease.

## Figures and Tables

**Figure 1 biosensors-11-00228-f001:**
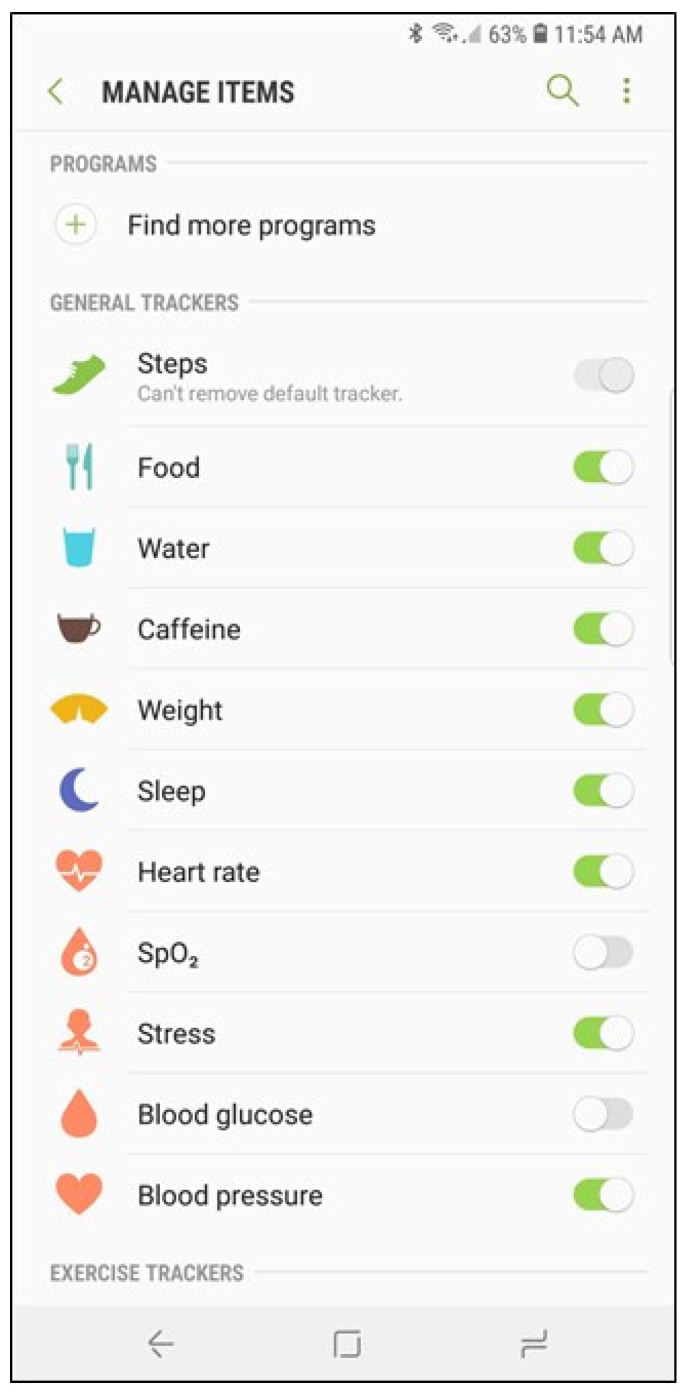
The screen of the Samsung Health Monitor application.

**Figure 2 biosensors-11-00228-f002:**
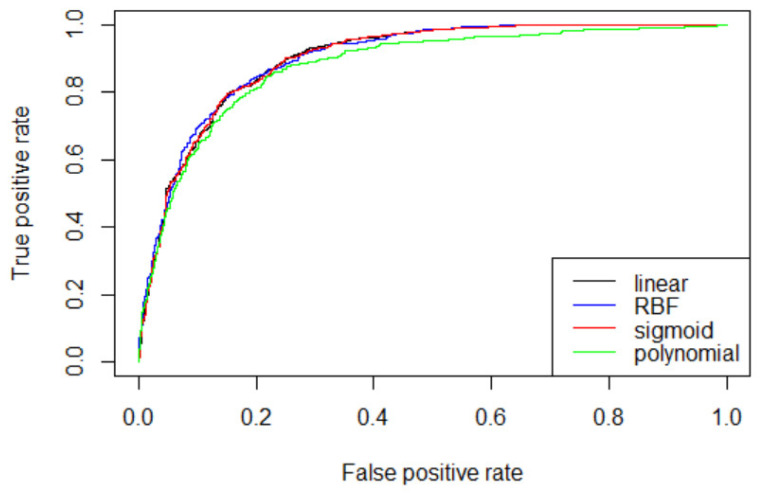
Comparison of ROC curve for four kernel functions.

**Figure 3 biosensors-11-00228-f003:**
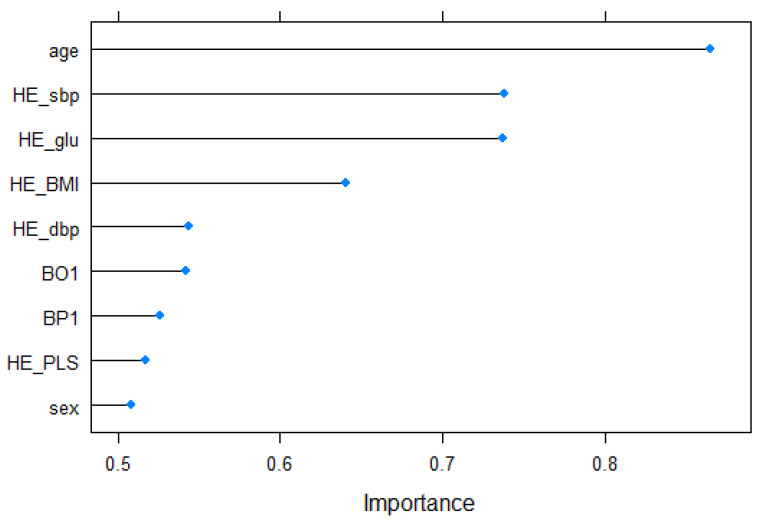
Features’ importance.

**Figure 4 biosensors-11-00228-f004:**
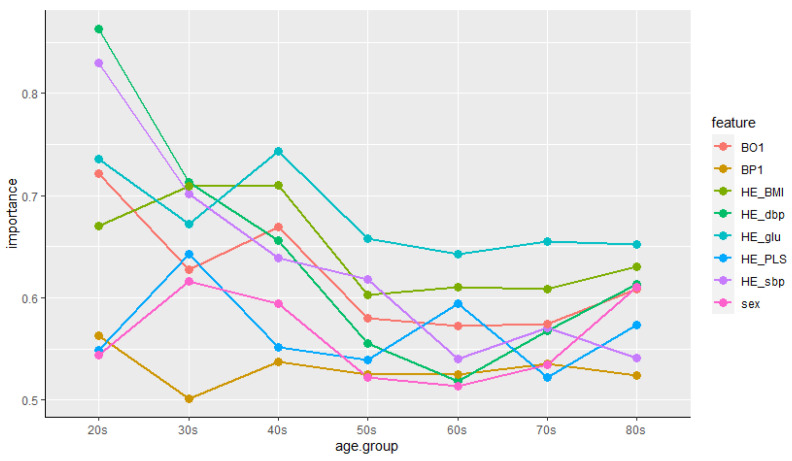
Features’ importance by age group.

**Table 1 biosensors-11-00228-t001:** Description of variables.

Variables	Variable Names	Description	Variable Types	Values of Measures {Min, Max}	Descriptive Statistics (Mean, SD)
target variable	chd_risk	prevalence of cardiovascular disease	nominal	0: no, 1: yes	no (4380), yes (1790)
input variables	sex	gender	nominal	1: male, 2: female	male (2722), female (3448)
	age	age	continuous	{10, 80}	(47.33, 19.45)
	HE_sbp	systolic blood pressure	continuous	{76, 215}	(118.21,16.08)
	HE_dbp	diastolic blood pressure	continuous	{34, 124}	(74.97, 10.09)
	HE_glu	fasting blood glucose	continuous	{35, 356}	(100.61, 22.97)
	HE_PLS	pulse rate for 15 s	continuous	{15, 29}	(17.80, 2.26)
	BP1	perceived stress	nominal	{1, 4}	(3.03, 1.24)
	BO1	self-awareness of body shape	nominal	{1, 5}	(3.36, 1.09)
	HE_BMI	body mass index	continuos	{13.50, 50.29}	(23.64, 3.78)

**Table 2 biosensors-11-00228-t002:** Descriptive statistics of variables of training and test data sets.

Variable Names	Training Data Set		Test Data Set	
	Mean, SD	1st qu., 3rd qu.	Mean, SD	1st qu., 3rd qu.
chd_risk	no (3099), yes (1220)		no (1281), yes (570)	
sex	male (1899), female (2420)		male (823), female (1028)	
age	(47.23, 19.47)	(33, 63)	(47.57, 19.42)	(33, 63)
HE_sbp	(118.1,16.03)	(106, 127)	(118.4,16.20)	(106, 128)
HE_dbp	(74.96, 10.10)	(69, 81)	(75, 10.07)	(69, 82)
HE_glu	(100.4, 22.45)	(89, 103)	(101.1, 24.13)	(90, 104)
HE_PLS	(17.79, 2.24)	(16, 19)	(17.82, 2.29)	(16, 19)
BP1	(3.04, 1.26)	(2, 3)	(3.02, 1.18)	(2, 3)
BO1	(3.37, 1.11)	(3, 4)	(3.35, 1.04)	(3, 4)
HE_BMI	(23.62, 3.78)	(21.07, 25.79)	(23.69, 3.76)	(21.04, 25.86)

**Table 3 biosensors-11-00228-t003:** Performance comparison of models on test data set.

Methods	Accuracy	Recall	Precision	F1 Score
logistic regression	82.55	85.59	89.93	87.70
artificial neural network	82.60	89.77	85.76	87.72
support vector machine	83.04	91.96	84.81	88.24

**Table 4 biosensors-11-00228-t004:** Accuracy and AUC for four kernel function with default values C and Gamma on test data set.

Kernel Functions	C	Gamma	Accuracy	AUC
Linear	1		82.55	0.781
Polynomial	1	0.111	80.93	0.727
RBF	1	0.111	83.04	0.775
Sigmoid	1	0.111	76.12	0.708

**Table 5 biosensors-11-00228-t005:** Accuracy and AUC of best performance for four kernel functions on test data set.

Kernel Functions	C	Gamma	Accuracy	AUC
Linear	1		82.55	0.781
Polynomial	100	0.1	81.96	0.761
RBF	10	0.01	83.20	0.778
Sigmoid	1	0.01	82.66	0.777

**Table 6 biosensors-11-00228-t006:** Features’ importance according to gender.

Male Features	Importance	Female Features	Importance
Age	0.8522	Age	0.8774
HE_glu	0.7336	HE_sbp	0.7897
HE_sbp	0.6648	HE_glu	0.7441
HE_BMI	0.6054	HE_BMI	0.6689
BO1	0.5495	HE_dbp	0.5801
BP1	0.5308	BO1	0.5368
HE_PLS	0.5218	BP1	0.5229
HE_dbp	0.5008	HE_PLS	0.5143

## Data Availability

The data presented in this study are available upon request from the corresponding author.

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
