# Peer review of "Building a Cardiovascular Disease Prediction Model for Smartwatch Users Using Machine Learning: Based on the Korea National Health and Nutrition Examination Survey"

_biosensors, 2021, doi:10.3390/bios11070228_

Round 1

Reviewer 1 Report

In this manuscript, the author aims to predict cardiovascular risk groups using machine learning by extracting data that can be measured with a smartwatch from the Korean National Health and Nutrition Examination Survey.
Although this research is beneficial to the Korean society, there are some issues regarding the machine learning model building method.

Comments
1) This title misleads readers into thinking that the prediction model was built based on data collected from smartwatches.
Please change the title appropriately.

2)Abstract
The authors should clearly state the sources of their information, the Korean National Health and Nutrition Examination Survey.

3)3.1.Data source
"The final 6,170 people were selected for analysis of this study after removing missing and abnormal values from the latest 2019 survey results."
Please specify the number of subjects in "The Korea National Health and Nutrition Examination Survey", the criteria for selection or exclusion of subjects in this study, and the definition of abnormal values.
Please illustrate each exclusion step as a scheme with the number of remaining subjects.

4)3.1.Data source
 "The reason for linking the prevalence of diabetes and the presence of cardiovascular disease is the result of research showing that if blood sugar continues to be high due to diabetes, blood vessels are damaged and various cardiovascular diseases are accompanied [23]."

Please add a note regarding the integration of dyslipidemia.

5)3.1.Data source
 "As mentioned above, Samsung Electronics' smartwatch can measure heart rate, oxygen saturation, stress index, blood pressure, and electrocardiogram As mentioned above, Samsung Electronics' smartwatch can measure heart rate, oxygen saturation, stress index, blood pressure, and electrocardiogram, and blood glucose can be measured in the second half of 2021."

It is unclear what "stress index" means. Please mention the content of this variable.

6) Table 1.
In order to use BP1 (perceived stress) as an explanatory variable in machine learning, it is necessary to prove that it corresponds to the stress index displayed by smartwatch. Please insert an explanation on this point in the text.

7) Please insert information about the details of each Variable shown in Table 1.
For example, please specify the number of men/women, the average and SD of blood pressure, etc.

8) 3.3. Research procedure and 4.1.Performance evaluation index
In this manuscript, there is no description of the validation method for the prediction model constructed by machine learning. In machine learning methods, overfitting must be avoided by implementing some kind of validation method. Since the sample size in this study is large enough, it is recommended to separate the test set group of about 20~30% from the training set group. After building the model with the training set, the prediction performance should be checked by the test set.

9) 4.3.1. Optimal model of support vector machine
Re-evaluate the model using the test set.

10) 4.3.2. features' importance
Describe the method for calculating the importance.

11) Conclusion
"In this study, predictions were made for all data 341 without dividing the data sets into training and test, but since the prediction results differ In this study, predictions were made for all data 341 without dividing the data sets into training and test, but since the prediction results differ depending on the training and test data sets, further research is needed on the minimum number of training data that does not affect prediction performance."

Avoiding overfitting is essential in machine learning.
A predictive model built without any validation is not a scientific achievement. Validation through the use of a test set is a minimum requirement for acceptance of this manuscript.

Author Response

In this manuscript, the author aims to predict cardiovascular risk groups using machine learning by extracting data that can be measured with a smartwatch from the Korean National Health and Nutrition Examination Survey. Although this research is beneficial to the Korean society, there are some issues regarding the machine learning model building method.

Comments

  • This title misleads readers into thinking that the prediction model was built based on data collected from smartwatches. Please change the title appropriately.
  • Following the reviewer’s comments, I have modified the title using ‘the Korea National Health and Nutrition Examination Survey’.
  • Abstract

The authors should clearly state the sources of their information, the Korean National Health and Nutrition Examination Survey.

  • As the reviewer commented, I have stated the sources of information, the Korea National Health and Nutrition Examination Survey to the abstract. Also, the Korea National Health and Nutrition Examination Survey was added to keywords.
  • 1. Data source

"The final 6,170 people were selected for analysis of this study after removing missing and abnormal values from the latest 2019 survey results." Please specify the number of subjects in "The Korea National Health and Nutrition Examination Survey", the criteria for selection or exclusion of subjects in this study, and the definition of abnormal values.

Please illustrate each exclusion step as a scheme with the number of remaining subjects.

  • Following the reviewer’s comments, I have explained the number of subjects in "The Korea National Health and Nutrition Examination Survey", the criteria for selection or exclusion of subjects in this study, and the definition of abnormal values in Section 3.1 (page 3).
  • 1.Data source

"The reason for linking the prevalence of diabetes and the presence of cardiovascular disease is the result of research showing that if blood sugar continues to be high due to diabetes, blood vessels are damaged and various cardiovascular diseases are accompanied [23]."

Please add a note regarding the integration of dyslipidemia.

  • Following the reviewer’s comments, I have added previous study on the relationship between dyslipidemia and cardiovascular disease in Section 3.2 (page 3).
  • 1.Data source

"As mentioned above, Samsung Electronics' smartwatch can measure heart rate, oxygen saturation, stress index, blood pressure, and electrocardiogram As mentioned above, Samsung Electronics' smartwatch can measure heart rate, oxygen saturation, stress index, blood pressure, and electrocardiogram, and blood glucose can be measured in the second half of 2021."

It is unclear what "stress index" means. Please mention the content of this variable.

  • To address the reviewer’s concerns, by dividing the sentence into two, the first sentence explained the aforementioned part, and the second sentence explained the already provided functions. Then I have added a description of the stress index in Section 3.2 (page 3-4).

6) Table 1.

In order to use BP1 (perceived stress) as an explanatory variable in machine learning, it is necessary to prove that it corresponds to the stress index displayed by smartwatch. Please insert an explanation on this point in the text.

  • To address the reviewer’s concerns, I have added that the subjective perceived stress level is used instead of the smartwatch's stress index because the perceived stress reflects the stress index displayed by smartwatch (page 4).

7) Please insert information about the details of each Variable shown in Table 1.
For example, please specify the number of men/women, the average and SD of blood pressure, etc.

  • Following the reviewer’s comments, I have added the details of each variable shown in Table 1 (page 5).

8) 3.3. Research procedure and 4.1.Performance evaluation index

In this manuscript, there is no description of the validation method for the prediction model constructed by machine learning. In machine learning methods, overfitting must be avoided by implementing some kind of validation method. Since the sample size in this study is large enough, it is recommended to separate the test set group of about 20~30% from the training set group. After building the model with the training set, the prediction performance should be checked by the test set.

  • To address the reviewer’s concerns, In order to avoid the problem of overfitting the model, about 70 % (4,319 respondents) of the observations in the entire data set are trained as a training data set. And the performance of the optimal model is verified with the test data set of 30% (1,851 respondents), which is the remaining data set not used for model training. I have explained the validation method for the prediction model in Section 3.3 (page 5)

9) 4.3.1. Optimal model of support vector machine

Re-evaluate the model using the test set.

  • Since support vector machine has the highest Accuracy, Recall, and F1 Score among the three models in the validation process for the prediction model in Section 4.2, I have re-evaluate the svm model using the test set in Section 4.3.

10) 4.3.2. features' importance

Describe the method for calculating the importance.

  • Following the reviewer’s comments, I have explained the method for calculating the importance in Section 4.3.2 (page 8). In addition, I have added the function name of R program.

11) Conclusion

"In this study, predictions were made for all data 341 without dividing the data sets into training and test, but since the prediction results differ In this study, predictions were made for all data 341 without dividing the data sets into training and test, but since the prediction results differ depending on the training and test data sets, further research is needed on the minimum number of training data that does not affect prediction performance."

Avoiding overfitting is essential in machine learning. A predictive model built without any validation is not a scientific achievement. Validation through the use of a test set is a minimum requirement for acceptance of this manuscript.

  • To address the reviewer’s concerns, In order to avoid the problem of overfitting the model, about 70 % (4,319 respondents) of the observations in the entire data set are trained as a training data set. And the performance of the optimal model is verified with the test data set of 30% (1,851 respondents), which is the remaining data set not used for model training. I have explained the validation method for the prediction model Then, since support vector machine has the highest Accuracy, Recall, and F1 Score among the three models in the validation process for the prediction model, I have re-evaluate the svm model using the test set.

I sincerely appreciate the reviewer’s thoughtful comments that helped me improve the paper.

Reviewer 2 Report

In "Building a Cardiovascular Disease Prediction Model for Smartwatch Users using Machine Learning" authors investigated the capability of three different classification models to predict either cardiovascular diseases or diabetes by utilizing as input data the information gathered by commercial smartwatches. Actually these devices are able to collect information about diastolic and systolic blood pressure, blood glucose, pulse rate, BMI. Furthermore two categorical parameters can be estimated: perceived stress and self-awareness of body shape.

The topic is surely interesting and worth of investigation. However in the reviewer opinion there are a series of points that should be very carefully considered by the authors in the methodological approach in order to have reliable results. In particular, here below the review listed a set of points that should be considered in the manuscript:

  • First of all, in order to avoid the overfitting of models (especially in case of non-linear classifiers) the dataset is randomly subdivided into training and test and the performances in test are evaluated. In order to provide statistics about the robustness of the model, many runs are performed considering each time different training/test combination ( each time selected randomly). This approach should be utilized, especially since the dataset comprises a large number of subjects. On the other hand, utilizing an unique dataset, it is difficult to evaluate the actual prediction capability.
  •   Secondly, the age seems to be the most important parameter, probably because the occurrence of cardiovascular diseases or diabetes is higher in elders. Due to this behavior, rather than including age in the parameters ( features) the samples are grouped by age intervals and different models are built on these different subsets to avoid bias on the distribution of classes due to age distributions of subjects. Similar observations can be made in the case of gender.

  • Thirdly in the reviewer opinion it is critical the way authors attributed the classes. They included in the "1" category all the people having at least 1 of the selected diseases. it would be surely useful to investigate the capability of discriminate the occurrence of each single disease. In the situation described by the authors, if a subject has one of these disease (e.g. diabetes) the model is not longer able to signal the arising of any of the other pathologies. It would much more significant to separate at least between diabetes and cardiovascular diseases occurrences. 

  • Finally, more details about distributions of parameters in the dataset should be reported ( e.g. the age, the gender distribution and occurrence of each pathology in the subjects of the dataset). Furthermore it should be indicated if the characteristic of sensors in the smartwatches allow ( or will allow) the monitoring of parameters with same resolution and Limits of detection of the parameters in the dataset.

Reviewer suggests to carefully consider the above points to provide a revised version of the manuscript. 

Author Response

In "Building a Cardiovascular Disease Prediction Model for Smartwatch Users using Machine Learning" authors investigated the capability of three different classification models to predict either cardiovascular diseases or diabetes by utilizing as input data the information gathered by commercial smartwatches. Actually these devices are able to collect information about diastolic and systolic blood pressure, blood glucose, pulse rate, BMI. Furthermore two categorical parameters can be estimated: perceived stress and self-awareness of body shape.

The topic is surely interesting and worth of investigation. However in the reviewer opinion there are a series of points that should be very carefully considered by the authors in the methodological approach in order to have reliable results. In particular, here below the review listed a set of points that should be considered in the manuscript:

First of all, in order to avoid the overfitting of models (especially in case of non-linear classifiers) the dataset is randomly subdivided into training and test and the performances in test are evaluated. In order to provide statistics about the robustness of the model, many runs are performed considering each time different training/test combination ( each time selected randomly). This approach should be utilized, especially since the dataset comprises a large number of subjects. On the other hand, utilizing an unique dataset, it is difficult to evaluate the actual prediction capability.

  • To address the reviewer’s concerns, In order to avoid the problem of overfitting the model, about 70 % (4,319 respondents) of the observations in the entire data set are trained as a training data set. And the performance of the optimal model is verified with the test data set of 30% (1,851 respondents), which is the remaining data set not used for model training. I have explained the validation method for the prediction model in Section 3.3. Then, since support vector machine has the highest Accuracy, Recall, and F1 Score among the three models in the validation process for the prediction model in Section 4.2, I have re-evaluate the svm model using the test set in Section 4.3.

Secondly, the age seems to be the most important parameter, probably because the occurrence of cardiovascular diseases or diabetes is higher in elders. Due to this behavior, rather than including age in the parameters ( features) the samples are grouped by age intervals and different models are built on these different subsets to avoid bias on the distribution of classes due to age distributions of subjects. Similar observations can be made in the case of gender.

  • Following the reviewer’s comments, I have additionally examined the features’ importance by age group and gender in Section 4.3.2. The features’ importance results by age group are summarized in a graph format, and the features’ importance results by gender are summarized in a table format. In addition, I have added the related previous studies.

Thirdly in the reviewer opinion it is critical the way authors attributed the classes. They included in the "1" category all the people having at least 1 of the selected diseases. it would be surely useful to investigate the capability of discriminate the occurrence of each single disease. In the situation described by the authors, if a subject has one of these disease (e.g. diabetes) the model is not longer able to signal the arising of any of the other pathologies. It would much more significant to separate at least between diabetes and cardiovascular diseases occurrences. 

  • As the reviewer is concerned, I think it's very important to distinguish between the characteristics of the diseases. However, it is different from the original purpose of this study, so I would like to proceed with further research, and I have stated this in the conclusion.

Finally, more details about distributions of parameters in the dataset should be reported ( e.g. the age, the gender distribution and occurrence of each pathology in the subjects of the dataset).

Furthermore it should be indicated if the characteristic of sensors in the smartwatches allow (or will allow) the monitoring of parameters with same resolution and Limits of detection of the parameters in the dataset. Reviewer suggests to carefully consider the above points to provide a revised version of the manuscript. 

  • Following the reviewer’s comments, I have added the details of each variable shown in Table 1 (5 page). The sensors of the smartwatch have already been verified at home and abroad as mentioned in paper, and the data from the Korea National Health and Nutrition Examination Survey is reliable data as approved statistics in Korea. These facts were the background of this study.

I sincerely appreciate the reviewer’s thoughtful comments that helped me improve the paper.

Round 2

Reviewer 1 Report

1.There was no evidence that the smartwatch stress index correlated with the subjective stress rating used in the study.
Reference # 25, cited by the author, contains a study on the correlation between pulse variation measured with a smartwatch and heart rate variation measured with a heartbeat measurement system. There is a description of the relationship between stress and heart rate variability, but there is no description of the relationship between subjective stress and stress measured in smartwatches.
As a limitation of this study, it should be stated that items related to stress are not sufficiently related and that the predictive model must be reconstructed from data from smartwatches.

2. Specify that the tables show the results of the test set.

3. The importance of BP1 from Fig4 is lower than sex and HE _ PLS in most ages.
This result does not appear to be consistent with Fig. 3. The authors should explain this point.

Author Response

  1. There was no evidence that the smartwatch stress index correlated with the subjective stress rating used in the study.

Reference # 25, cited by the author, contains a study on the correlation between pulse variation measured with a smartwatch and heart rate variation measured with a heartbeat measurement system. There is a description of the relationship between stress and heart rate variability, but there is no description of the relationship between subjective stress and stress measured in smartwatches.

As a limitation of this study, it should be stated that items related to stress are not sufficiently related and that the predictive model must be reconstructed from data from smartwatches.

  • To address the reviewer’s concerns, other previous research was found and cited, and nonetheless, a limitation of this study is described in page 4-5. I have also added on the limitation of the data variables in the Conclusions section.
  1. Specify that the tables show the results of the test set.
  • Following the reviewer’s comments, I have explained this in the first sentence of Section 4.2 and specified in Tables 3, 4 and 5.
  1. The importance of BP1 from Fig4 is lower than sex and HE _ PLS in most ages.

This result does not appear to be consistent with Fig. 3. The authors should explain this point.

  • As the reviewer commented, I have explained this point and related previous studies and added a limitation of the perceived stress variable in page 9.

I sincerely appreciate the reviewer’s detailed and thoughtful comments that helped me improve the paper.

Reviewer 2 Report

The reviewer appreciates the efforts made by the authors to improve the manuscript by following the remarks made during the reviewing process. In the reviewer opinion the scientific soundness of manuscript is greatly improved. In the reviewer opinion there are still some points that can/should be amended to improved the comprehension of the manuscript.

  • Statistical data should be reported separately for training and test. Futhermore since they probably will not obey to a Gaussian distribution, it should be more correct to report the distribution in terms of mean and percentiles (e.g. 25th and 75th).
  • The data utilized by the authors are acquired by professional instruments. However the author proposes the implementing of smartwatches. At the actual state of the art, have these instrument  resolution,limit of detection and reproducibility comparable of those utilized for acquiring the data in the dataset? In the author opinion, can this influence the classification results?
  • Author state that classification on single classes will be performed in the future. Indeed in this paper it would be interesting to understand if the false negative (FN) subjects are well distributed among the different diseases/classes or if the models misclassify more frequently the subjects having a particular diseases.

Author Response

The reviewer appreciates the efforts made by the authors to improve the manuscript by following the remarks made during the reviewing process. In the reviewer opinion the scientific soundness of manuscript is greatly improved. In the reviewer opinion there are still some points that can/should be amended to improved the comprehension of the manuscript.

Statistical data should be reported separately for training and test. Futhermore since they probably will not obey to a Gaussian distribution, it should be more correct to report the distribution in terms of mean and percentiles (e.g. 25th and 75th).

  • Following the reviewer’s comments, I have summarized descriptive statistics for each variable of the training and the test data set in Table 2 of page 6.

The data utilized by the authors are acquired by professional instruments. However the author proposes the implementing of smartwatches. At the actual state of the art, have these instrument resolutions, limit of detection and reproducibility comparable of those utilized for acquiring the data in the dataset? In the author opinion, can this influence the classification results?

  • To address the reviewer’s concerns, in the Conclusion section, I have explained the difference between the data I used in this study and the data implemented on the smartwatch. In addition, I have stated the stress index, which exist differences between the data I used in this study and the data implemented on the smartwatch.

Author state that classification on single classes will be performed in the future. Indeed in this paper it would be interesting to understand if the false negative (FN) subjects are well distributed among the different diseases/classes or if the models misclassify more frequently the subjects having a particular disease.

  • As the reviewer commented, I have added the contribution of future research in Conclusion section.

I sincerely appreciate the reviewer’s detailed and thoughtful comments that helped me improve the paper.

Round 3

Reviewer 1 Report

The author responded appropriately to all comments.